# Observational Scaling Laws in LLM-based Embodied Decision Making

## Abstract

We introduce an observational method to derive scaling laws for LLM performance in embodied decision-making tasks, allowing us to predict embodied skills, quantify simulation gaps, and algorithm intervention. In contrast to conventional scaling research that trains multiple models from scratch at different scales, our approach bypasses new model training and instead uses publicly available pre-trained LLMs to model performance trends across different model families and sizes. Constructing such unified scaling law across diverse model families is challenging, as these models differ in both training compute efficiency and resulting capabilities. We address this by employing a generalized scaling framework that expresses model performance as a function of a low-dimensional capability space. We first validate such scaling law on the Embodied Agent Interface (EAI) benchmark across 125 LLMs, confirming a predictive accuracy that represents at least a 50% improvement over traditional compute scaling laws. We then find that an LLM's decision-making ability is highly predictable—accurately forecasting the performance of larger models using data from those as small as 40B parameters—which allows us to quantify both the performance gap between simulation environments and the impact of structured decoding.

## 1 Introduction

Large Language Models (LLMs) power embodied agents that interpret goals, plan actions, and interact with dynamic environments. This progress presents a fundamental question: do classical scaling laws, which link model size to performance on language benchmarks, hold true for embodied tasks? These complex tasks demand structured action sequences and sequential decisions (1; 2), a clear departure from simple text generation. Answering this question is critical to predict the capabilities of larger models and to guide the future development of embodied AI.

Prior work establishes systematic benchmarks to evaluate LLMs in embodied settings, measuring capabilities like goal interpretation, subgoal decomposition, and action sequencing (3). These benchmarks reveal what current models can do, but they do not explain how performance scales. A clear relationship between compute, such as training FLOPs, and downstream embodied success remains unestablished. This disconnect is critical. Embodied tasks introduce challenges like environmental interaction and the sim-to-real gap, which are absent from the pure language domains where scaling laws are well-understood (4). The field currently lacks predictive models for embodied AI performance as a function of model scale, leaving a fundamental gap between capability evaluation and scaling science.

To move beyond evaluating the performance of individual models and toward a systematic study of their scaling behavior, we introduce an observational scaling approach to predict the performance of LLMs in embodied tasks. Our method extends observational scaling laws (5), which use a low-dimensional representation of a model's capabilities to forecast success on complex downstream tasks. This technique allows us to build predictive scaling models without the need for costly, full-scale retraining, bridging the gap between embodied AI evaluation and scaling science.

Our work starts with the observation of general scaling laws across LM families that relate downstream performance to training measures. We test if this relationship extends to complex embodied skills. An agent's success in goal interpretation, for example, understanding "bring me the red cup

from the kitchen," depends on core skills like natural language understanding and commonsense reasoning. We posit that a model's downstream performance is a function of a low-dimensional space of such capabilities. Model families differ only in the efficiency with which they convert training compute into these capabilities. This relationship implies a log-linear trend from capabilities to downstream performance across all model families, and a log-linear trend from training compute to capabilities within each specific family.

This observational approach provides key advantages. First, it enables the study of scaling behavior without retraining models. Second, it combines models from heterogeneous families with different scaling properties, such as LLaMA (6; 7; 8), Qwen(9; 10; 11; 12), Gemma(13; 14; 15; 16; 17), and StarCoder (18; 19). This allows an analysis of different scaling strategies and their impact on downstream performance and algorithmic interventions.

In experiments, we validate these scaling laws on the Embodied Agent Interface (EAI) benchmark using 125 open LLMs from 28 model families. We demonstrate our method's utility in three settings: predicting emergent capabilities, quantifying simulation gaps, and measuring the effect of structured outputs. First, we predict the performance of models larger than 40B parameters using data from models smaller than 40B. Second, we use the scaling laws to quantify the performance gap between different simulation environments. Third, we quantify the effect of structured outputs and find they degrade the model's decision-making performance.

Our contributions are twofold. First, we introduce an observational scaling framework that unifies scaling laws for embodied tasks. This framework predicts decision-making performance as a function of model capabilities and scale. Second, using our framework on the EAI benchmark, we quantify the performance degradation from structured outputs and measure the gap between simulation environments.

The paper proceeds as follows. Section 2 reviews related work. Section 3 formulates our problem. Section 4 presents our method and Section 5 details our experiments. We conclude in Section 6.

## 2 RELATED WORKS

**Embodied benchmarks** such as VirtualHome, ALFRED, BEHAVIOR-1K, TEACh, and Habitat evaluate whether agents can map goals and observations into machine-executable action–state sequences that achieve task goals, enabling step- and goal-level verification of decision making (20; 21; 22; 23; 24). The Embodied Agent Interface (EAI) formalizes this setting by standardizing four LLM decision-making modules (goal interpretation, subgoal decomposition, action sequencing, transition modeling), specifying I/O formats, and adding fine-grained error taxonomies that support modular, diagnostic evaluation (3). Building on these interfaces, researchers either (i) improve LLM performance via prompting and planning—e.g., Chain-of-Thought and ReAct—or affordance-grounded planning for robotics (SayCan), or (ii) extend toward VLA policies that couple vision, language, and action for robot control (RT-2; OpenVLA) (25; 26; 27; 28; 29; 30; 31; 32; 33; 34; 35; 36). Unlike work that augments algorithms or expands tasks, our focus is to analyze scaling behavior of LLMs within EAI, linking standardized upstream capabilities (reasoning, coding, math) to downstream embodied performance via observational scaling principles (5).

**Scaling laws** fall into two main categories: compute-based scaling laws and downstream performance scaling laws. Standard scaling laws (37; 38; 39; 40; 41; 4; 42), which are compute-based scaling laws, are typically expressed as power-law relationships between a model's cross-entropy loss $L$ and compute-scale measures. In this context, "compute scale" refers to training resources such as the number of training FLOPs ($C$), model parameters ($N$), and training tokens ($D$). Compute-based scaling laws characterize pretraining behavior within a single model family, linking upstream performance to controllable quantities like training compute. In contrast, downstream performance scaling laws (43; 39; 44; 45; 46; 38) analyze scaling across model families, connecting benchmark results to compute-related metrics (e.g., model size (47)) or predicting its performance due to appearing rapid "emergence" (48; 49; 50). Specifically, Researchers (51; 52) have explored both linear and sigmoidal functional forms to extrapolate downstream performance from pretraining loss or compute measures. Chen et al. (53) introduced a two-stage approach—first predicting pretraining loss from compute, then mapping that loss to downstream performance—even when using models from different families with varying compute-efficiencies. On the theory front, Arora and Goyal (54) and Ruan

et al. (5) derive theories characterizing how performance on complex skills of LMs can be derived as a composition of base skills. Drawing from these downstream scaling insights including observation scaling laws (5), we aim to identify scaling patterns between embodied decision-making performance and conventional benchmark metrics, emphasizing empirical observation rather than compute-driven modeling.

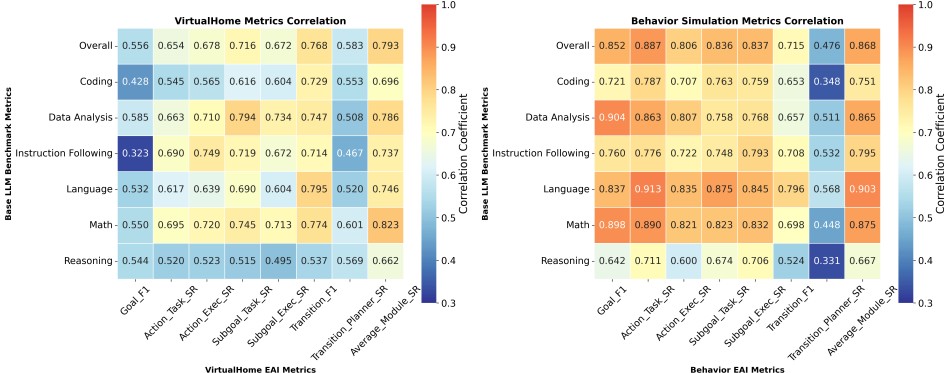

Figure 1: Pearson correlation heatmap between LiveBench 08-31-2024 and EAI task metrics.

**Corelation between benchmarks** have been investigated in numerous works. Specifically, extensive research has explored the relationship between the out-of-distribution performance and in-distribution performance of machine learning models (55; 56; 57; 58; 59). In NLP and LM evaluations, Qiu et al. (60) and Torregrossa et al. (61) found that multiple evaluation metrics for word embeddings are highly correlated, while Liu et al. (62) observed robust correlations across question-answering benchmarks. Perlitz et al. (63) and Polo et al. (64) further noted that performance is strongly correlated across samples of different LM benchmarks, enabling the design of more efficient evaluation suites. Beyond empirical observations, studies have identified compact latent structures driving performance across tasks. Ilić (65) demonstrated that a single latent factor accounts for 85% of performance variance on the Open LLM Leaderboard (66) and GLUE benchmark (67). Burnell et al. (68) similarly uncovered that three factors explain 82% of variation on the HELM benchmark (69). These findings align with cross-task consistency seen elsewhere: for example, MixEval (70) combines diverse benchmark queries and achieves a high ranking correlation (Pearson 0.96) with human-composed Chatbot Arena (71), demonstrating coherence between aggregated benchmarks and human judgment. In Figure 1, our work observes benchmark correlations (the highest is 91.3%) between LiveBench(72) and EAI, and finds the gap between simulations. This leads to formulating simulation-aware scaling predictions based on benchmark performance.

## 3 PROBLEM FORMULATION

We formulate scaling laws within the Embodied Agent Interface (EAI) benchmark (3). Our objective is to determine if a smooth scaling relationship exists between general-purpose LLM capabilities and the specialized skills of Goal Interpretation and Action Sequencing.

### 3.1 EMBODIED AGENT INTERFACE

EAI is a benchmark for embodied decision-making. It uses Linear Temporal Logic (LTL) as a formal language to represent goals and plans, enabling a precise evaluation of an agent's ability to understand instructions and generate action sequences. For a comprehensive overview of the framework's formalisms, including its state representation and LTL semantics, we refer the reader to the original EAI paper. In this work, we focus specifically on evaluating the Goal Interpretation and Action Sequencing modules.

Goal Interpretation module $\mathcal{G}$ translates a natural language instruction $l_g$ into a formal LTL goal $g$, given an initial state $s_0$. The **Input-Output** is $\mathcal{G} : (s_0, l_g) \rightarrow g$. Its performance is measured by an $F_1$ **set-matching score** between the generated goal $\hat{g}$ and the ground truth.

Action Sequencing module $Q$ generates an action sequence $\bar{a}$ to achieve a given LTL goal $g$ from a state $s_0$. The **Input-Output** is $Q : (s_0, g) \rightarrow \bar{a}$. It is evaluated on two metrics: **Trajectory**

**Feasibility** (whether the sequence $\bar{a}$ is executable in a simulator) and **Goal Satisfaction** (whether the resulting trajectory achieves $g$).

## 3.2 PROBLEM FORMULATION

Let $E_m$ be the normalized performance metric for a given model $m$ on a specific EAI evaluation task. We focus on key indicators of embodied competence, such as the `task_success_rate` or `execution_success_rate` for the Action Sequencing module, and the `all_f1` score for the Goal Interpretation module.

Let $C_m$ be the model's $m$ training compute in FLOPs, $N_m$ be parameter size, and $D_m$ be the pretraining token size. Following Kaplan et al. (4), we estimate $C_m$ using the approximation $C_m \approx 6N_m D_m$. This allows us to connect the concrete properties of a model to its performance.

Recent studies (51; 52) have found that a predictable scaling relationship holds for models within a single architectural family (e.g., Llama, Gemma, or Qwen). They observe a sigmoidal relationship between training compute and task error, formally expressed using a generalized linear model with a logistic link function ($\sigma^{-1}$):

$$\sigma^{-1}(E_m) \approx \lambda_f \log(C_m) + \mu_f. \tag{1}$$

Here, $\lambda_f$ and $\mu_f$ are constants that are determined empirically.

Our primary goal is to generalize this relationship to better quantify the scaling laws of the EAI benchmark, potentially finding a more universal framework that holds across different model families. A successful generalization would allow for more robust performance forecasting.

## 4 METHOD: OBSERVATIONAL SCALING LAWS

Our work builds on the Obscaling (5), a framework for creating a universal scaling model for diverse language models, including those with unknown training compute. This approach allows us to move beyond the family-specific limitations discussed previously and pursue a more generalizable law.

**Hypothesis 1 (Universal Performance Model)** The core hypothesis is that we can predict a model's ($m$) performance on a complex task (measured by error $E_m \in \mathbb{R}$) using a universal linear model based on a latent low-dimensional capability vector $S_m \in \mathbb{R}^K$:

$$\sigma^{-1}(E_m) \approx \beta^\top S_m + \alpha. \tag{2}$$

Here, $S_m$ is the capability vector for model $m$ in a $K$-dimensional space, $\sigma$ is the logistic function, $\beta \in \mathbb{R}^K$ is a universal weight vector that maps capabilities to performance, and $\alpha \in \mathbb{R}$ is a scalar bias.

**Hypothesis 2 (Latent Capability Projection)** Then, we hypothesize that a model's latent capability, $S_m$, is a linear projection of its benchmark performance vector, $B_m \in \mathbb{R}^T$. To compute this capability vector, we apply a projection matrix $\gamma \in \mathbb{R}^{K \times T}$ such that

$$S_m := \gamma B_m. \tag{3}$$

We derive this matrix by applying Principal Component Analysis (PCA) to the performance vectors of all models. The rows of $\gamma$ consist of the top $K$ principal components, as exemplified in Figure 2b.

**Hypothesis 3 (Log-linear Capability Scaling)** Next, if we assume that within a specific model family $f$, capability grows log-linearly with compute (Equation 4), then substituting this into Equation 2 recovers the familiar family-specific scaling law (Equation 1):

$$S_m \approx \theta_f \log(C_m) + \nu_f \tag{4}$$

$$\sigma^{-1}(E_m) \approx w_f \log(C_m) + b_f. \tag{5}$$

Here, $\theta_f \in \mathbb{R}^K$ is a family-specific vector, $\nu_f \in \mathbb{R}^K$ is a bias vector, $w_f = \beta^\top \theta_f$ and $b_f = \beta^\top \nu_f + \alpha$. Thus, Equation 5 is consistent with Equation 1

**Fitting Observational Scaling Laws** We begin with a set of LMs $\mathcal{M}$, and four quantities for each model $m \in \mathcal{M}$: its compute measure FLOPs $C_m$, its vector of benchmark scores $B_m$, and its

performance on a complex task $E_m$. From this data, we estimate the scaling relationship through a multi-stage procedure.

Firstly, we estimate the capability vectors $S_m$ via fitting PCA on $B_m$, and then find the universal parameters $\beta^*$ and $\alpha^*$ by minimizing the squared error for the relationship defined in Equation 2:

$$(h^*, \beta^*, \alpha^*) = \underset{h,\beta,\alpha}{\operatorname{argmin}} \sum_{m \in \mathcal{M}} \|(E_m) - h\sigma^{(}\beta^\top S_m + \alpha)\|^2. \tag{6}$$

where $\beta \in \mathbb{R}^K$, $\alpha \in \mathbb{R}$ are regression weights and bias. $h \in [0, 1]$ is the sigmoid scale and it results in $h^* = 1$ in most experiments. This defines a scalar capability score $P_m := (\beta^*)^\top \hat{S}_m + \alpha^*$ for any model.

Secondly, we determine the coefficients $w_f^*$ and $b_f^*$ for the scaling law described in Equation 5. Specifically, we select a reference family (e.g. llama-2) $f$, and then fit another linear regression using only the models from the reference family $f$:

$$(w_f^*, b_f^*) = \underset{w_f, b_f}{\operatorname{argmin}} \sum_{m \in f} \|P_m - (w_f \log(C_m) + b_f)\|^2, \tag{7}$$

This mapping allows us to convert any model's capability score $P_m$ into an intuitive metric—the $f$-equivalent FLOPs, $\tilde{C}_{m,f}$—by inverting the relation: $\log(\tilde{C}_{m,f}) := (P_m - b_f^*)/w_f^*$. This provides a single, compute-anchored axis for comparing all models.

## 5 EXPERIMENTS

Our experimental evaluation proceeds in four stages. First, we verify the core assumptions underlying our proposed observational scaling laws. Second, we validate the laws by fitting them to LLM performance on the EAI benchmark. Third, we apply the validated laws to identify a "simulation gap" between distinct EAI environments. Finally, we demonstrate the practical utility of our findings for model intervention by quantifying the performance impact of structured decoding.

**Experimental setup** We evaluate 124 open-source LLMs (listed in Tables 1 and 2) using the llama-factory (73) with vLLM backend (74) for efficient inference. We measure the target performance metric ($E_m$) on two tasks from the EAI benchmark (3): Action Sequencing (task and execution success rates) and Goal Interpretation (F1 score). To establish a general capability measure ($S$), we gather scores from the OpenLLM Leaderboard (66) on benchmarks testing reasoning (e.g., BBH, MATH), instruction following (IFEval), and expert knowledge (e.g., MMLU-PRO). We then apply Principal Component Analysis (PCA) with $K = 3$ to these general scores to derive our final capability measure $S$. We release the result and code in the supplementary materials.

### 5.1 VALIATION OF ASSUMPTION ON OBSERVATION SCALING LAWS

We validate two assumptions including Hypothesis 0 and Hypothesis 1, since we use different metrics than the original paper (5).

**Hypothesis 0 and 1** posit that a low-dimensional latent variable can effectively represent model performance. To extract this variable, we apply Principal Component Analysis (PCA with $K = 5$) to the full suite of benchmark metrics ($B$). We define the resulting components as the "principal capability" (PC) measures, $S$ (see (5) for additional details).

Our analysis validates this low-rank assumption. As shown in Figure 2a, the top three PCs capture approximately 97% of the total variance, with the first PC alone accounting for nearly 70%. Furthermore, these PCs are highly interpretable (Figure 2b). **PC-1** represents a **"general capability"**, **PC-2** corresponds to **"instruction following"**, and **PC-3** reflects **"mathematical reasoning"**. This evidence indicates that the complex LM capabilities covered by our benchmarks can be expressed as a linear combination of a few fundamental principal capabilities $S$.

**Hypothesis 3** proposes a linear relationship between the principal capability measures ($S$) and log-scale training compute ($C$). We use the first principal component (**PC-1**) to represent a model's capability $S$. We estimate the training compute $C$ by collecting the model parameter count ($N$) and

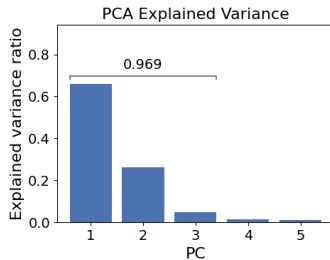 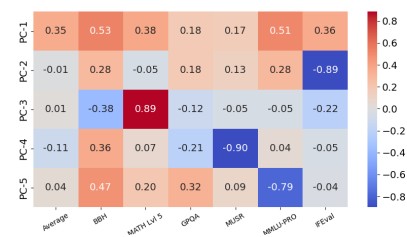

(a) PCA explained variance      (b) Principle component weights

Figure 2: Just a few capability dimensions explain most variability on a diverse range of standard LM benchmarks. We find that (a) the benchmark-model matrix is low-dimensional with the top 3 PCs explaining $\sim 97\%$ of the variance and (b) the PCs are interpretable: PC-1, PC-2, and PC-3 emphasize LMs' general, instruction-following, mathematical reasoning capabilities, respectively.

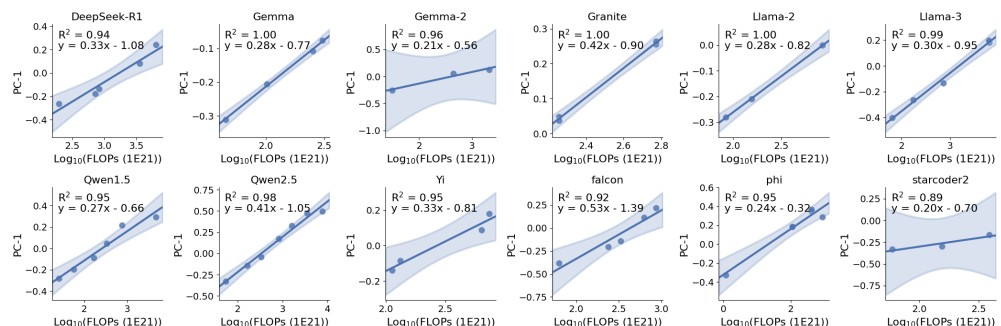

Figure 3: The extracted PC measures linearly correlate with log-compute within each model family. The linearity generally holds for various model families.

pretraining token size ($D$) from technical reports and project pages, then approximating the training FLOPs as $C \approx 6ND$.

Figure 3 validates this log-linear relationship within specific model families. We find a strong correlation where the PC-1 measure scales linearly with log-training FLOPs, achieving an $R^2 > 0.89$. This trend holds across diverse model architectures, including distilled models like DeepSeek-R1 (75) and code-focused models like StarCoder2 (19). The relationship also extends to lower-ranked components such as PC-2 and PC-3 (Figures 9 and 10). This empirical evidence supports our hypothesis in Equations 3 and 4, which state that different model families convert compute into capabilities at varying efficiencies within a shared capability space.

## 5.2 VALIDATING OBSERVATIONAL SCALING LAWS

Our objective is to validate that observational scaling laws can predict the performance of large language models on EAI tasks

**Experiment setup** We filter the dataset to ensure quality, excluding models with (1) zero task performance (e.g., max token length < input length, indicating evaluation failure) or (2) missing benchmark scores. To test extrapolation, we split models by size: those with <40B parameters form the training set, and larger ones the test set. All preprocessing steps and scaling-law parameters (Equations 7, 6) are fitted on the training data and applied to the test set to avoid information leakage.

**Baselines**: We compare the observational scaling law against two baselines, fitted by Eqation 1: (i)Model Size Scaling: A power-law fit based on the number of model parameters. (ii) Training FLOPs Scaling: A power-law fit based on the estimated floating-point operations used for training.

We present the results in Figure 4, and Figure 13, 14, 15 (appendix). Our method achieves the lowest Mean Squared Error (MSE) on the held-out test set of models with $\geq 40B$ parameters. For instance, on the Task Success Rate (Behavior) metric, the observational law yields a test MSE of $1.3 \times 10^{-3}$. This is more than an order of magnitude better than both the Model Size baseline ($6.5 \times 10^{-2}$) and the Training FLOPs baseline ($5.6 \times 10^{-2}$). A similar trend holds for the Execution Success Rate (Behavior) task.

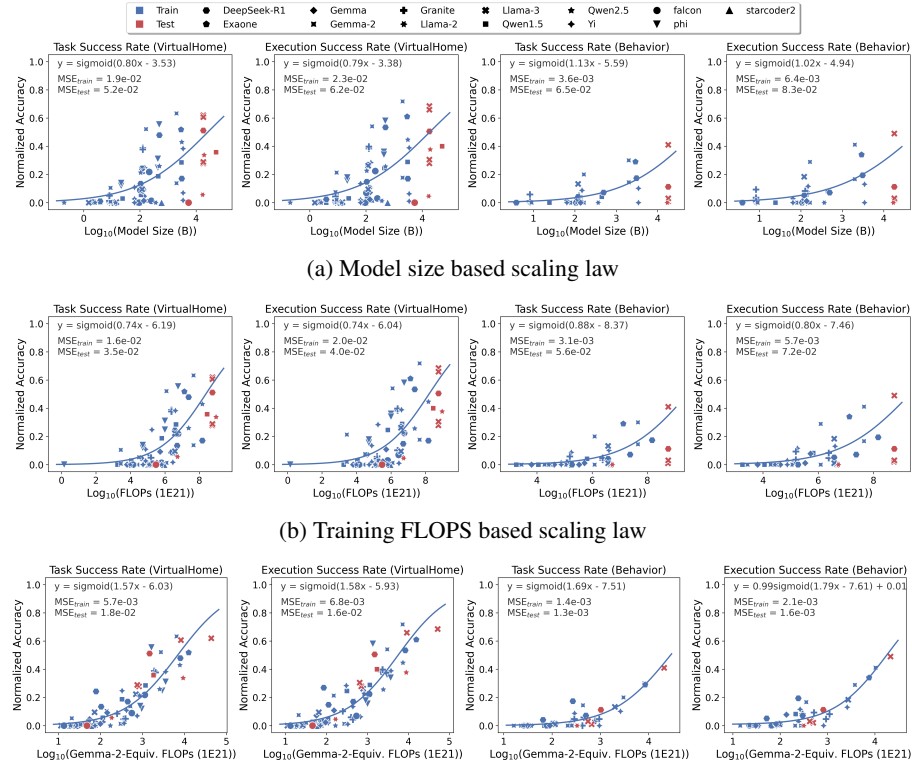

(a) Model size based scaling law

(b) Training FLOPS based scaling law

(c) Observational scaling law

Figure 4: Scaling curves of action sequencing on Virtualhome and Behavior. We compare three scaling laws: (a) Model Size, (b) Training FLOPs, and (c) our proposed Observational scaling law. Each plot shows the training data ($< 40B$ parameters, blue circles), the held-out test data ($\geq 40B$ parameters, red crosses), and the fitted sigmoid curve. The reported Mean Squared Error (MSE) on the train and test sets shows that the observational scaling law consistently achieves the lowest test MSE, indicating its superior ability to extrapolate performance to larger, more capable models. The fitted sigmoid curve is expressed as $y = sigmoid(n_1 \times n_2)$ where the coefficients $n_1$, $n_2$ corresponds to the regression weight $w_f^*$ and the bias $b_f*$ in Equation 5.

## 5.3 QUANTIFYING SCALING GAP BETWEEN SIMULATIONS

**Interpreting Scaling Law Coefficients** Following the validation in Section 5.2, we analyze the fitted regression coefficients from our observational scaling law to understand the relative difficulty of the tasks in different simluations.

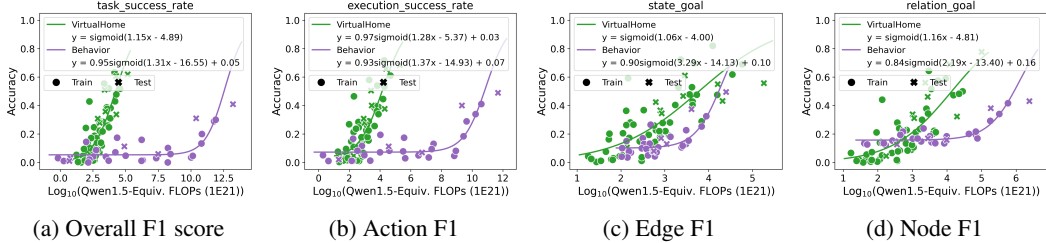

(a) Overall F1 score  (b) Action F1  (c) Edge F1  (d) Node F1

Figure 5: Comparison of observational scaling laws for Virtualhome and behavior on the action sequencing task.

From Figure 5, we observe two key trends: (i) For both the `Virtualhome` and `Behavior` environments, the regression weight ($w_f^*$) for `execution success rate` is consistently larger than the weight for `task success rate`. This is consistent with the logical constraint that a successful task execution is a stricter, and therefore more difficult, condition to satisfy than a plan that is merely executable. (ii) When comparing the two environments for the same task, `Behavior` exhibits a higher regression weight ($w_f^*$) but a lower bias ($b_f^*$) than `Virtualhome`. This suggests

that the `Behavior` simulation presents a higher initial barrier to effective performance (lower bias), but that performance scales more steeply with increasing model capability (higher weight) once a baseline of competence is achieved.

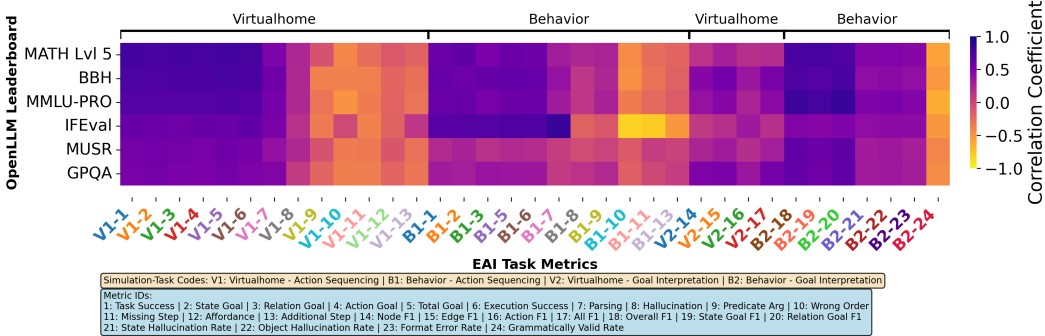

Figure 6: Correlation heatmap between EAI task metrics (x-axis) and base LLM benchmarks from the OpenLLM Leaderboard (y-axis). Dark purple cells indicate positive correlation, and light yellow cells indicate negative correlation. The plot reveals that different EAI tasks and environments draw on different foundational capabilities, such as mathematical reasoning (`MATH Lvl 5`) for `Virtualhome` and instruction following (`IFEval`) for `Behavior`.

**Correlation with Foundational LLM Capabilities** To contextualize the skills required by our EAI benchmarks, we compute the correlation between EAI task performance and scores from the Open-LLM Leaderboard. We present the results in the heatmap in Figure 6 and statistics in Table 3, 5, 4, 6. The analysis reveals that task performance in different simulation environments is associated with distinct underlying LLM capabilities. Specifically, Action Sequencing performance in `Virtualhome` shows a strong positive correlation with mathematical reasoning benchmarks (`MATH Lvl 5`). In contrast, the same task in the `Behavior` environment correlates most strongly with instruction following capabilities (`IFEval`). This suggests that `Virtualhome` may test a model's logical planning and reasoning abilities more heavily, while `Behavior` emphasizes the precise interpretation and execution of commands. Furthermore, we note that specific error metrics within our benchmark, such as `Missing Step` (V1-11) and `Affordance` (V1-12), show weak to no correlation with any of the general OpenLLM benchmarks. This indicates that standard LLM evaluations do not adequately measure a model's proficiency in these crucial aspects of embodied planning, highlighting a potential gap in existing evaluation practices.

## 5.4 THE IMPACT OF STRUCTURED DECODING

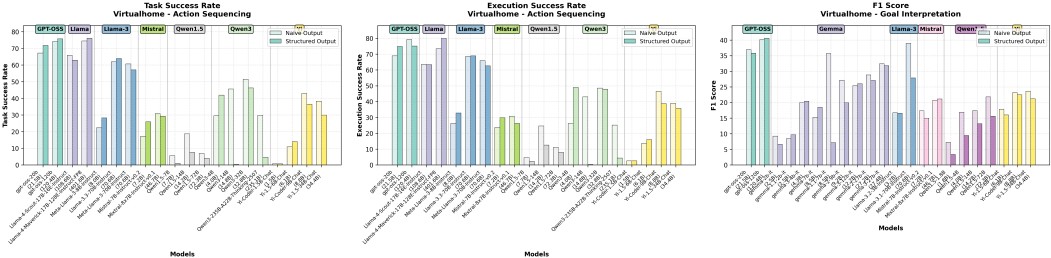

Figure 7: Comparison of observational scaling laws for standard generation (Base Model) versus structured decoding (Model with Decoder Masking) on the Virtualhome goal interpretation task. The plots show performance on action sequencing and goal interpretation tasks on Virtualhome.

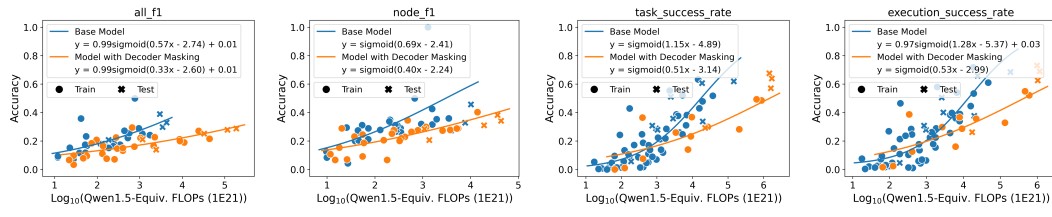

| (a) Overall F1 score | (b) Node F1 | (c) Overall F1 score | (d) Action F1 |

Figure 8: Comparison of observational scaling laws for standard generation (Base Model) versus structured decoding (Model with Decoder Masking). The y axis of first two plot represents metrics of Virtualhome's goal interpretation task and the last two are the metrics of Virtualhome's action sequencing task. Blue lines represent the base model and yellow represent the Model with Decoder Masking

In this section, we investigate the impact of enforcing structured outputs on the performance of LLMs in EAI tasks. While compelling models to generate plans in a specific format like JSON guarantees syntactic correctness and eliminates parsing failures, it is unclear whether this constraint helps or hinders the model's underlying reasoning and planning capabilities.

To quantify this trade-off, we designed a controlled experiment to measure performance differences between standard and constrained decoding. Using vLLM (74) as our inference backend, we evaluated a suite of models on our EAI benchmarks under two conditions: (1) with standard, unconstrained text generation, and (2) with structured decoding enabled via Xgrammar(76) to enforce a strict JSON output schema. The performance in both conditions was measured using the primary success metrics from our benchmark to isolate the effect of the decoding constraint. We report results in Figures 8,12.

A direct model-by-model comparison in Figure 11 reveals that the impact of structured decoding is not uniform and can be difficult to predict. For instance, on the `Action Sequencing` task, the constraint improves the `Task Success Rate` for capable models like `Llama-3-70B`, but it harms the performance of others like `Yi-1.5-6B`. Similarly, for `Goal Interpretation`, structured decoding hurts the performance of `GPT-4` and `Mixtral-8x7B`, yet provides a notable benefit to models such as `Yi-1.5-34B` and `Phi-3-mini-128k`.

As shown in Figure 8, our results for the Virtualhome goal interpretation task reveal that forcing a structured output consistently hurts model performance. For the main `Overall F1` score, models with the output constraint always performed slightly worse than the regular models, even though both improved at a similar rate as they scaled up. This performance gap was much larger on more detailed sub-tasks. For example, on `Edge classification F1`, the regular model's performance improved more than four times faster than the constrained model's (a scaling slope of w=0.79 vs. w=0.19). This suggests that while structured decoding guarantees a clean output format, these strict rules prevent the model from fully learning the complex relationships needed for the planning task.

## 6 CONCLUSION

This paper presents an observational method for deriving scaling laws in embodied decision-making, leveraging a large set of public LLMs to avoid costly training. Our generalized scaling law maps performance to a low-dimensional capability space, effectively modeling diverse model families. Validated on the EAI benchmark, our method shows high predictive accuracy, significantly improving on traditional compute-based laws. This framework provides a cost-effective way to forecast model performance, quantify the effects of interventions like structured decoding, and measure simulation gaps. Future work can extend this approach to build a unified model of the simulation gap and to quantify the effects of more complex LM interventions.

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

# A APPENDIX

## A.1 EXPERIMENT MODEL INFORMATION

## A.2 PC MEASURES LINEARLY CORRELATED WITH LOG-COMPUTE MEASURES

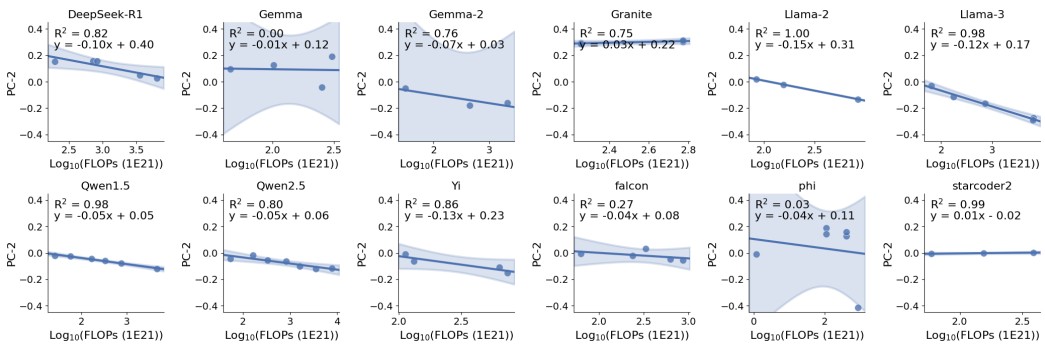

Figure 9: Linear relationship between the second principal component (PC-2) and log-compute across different model families.

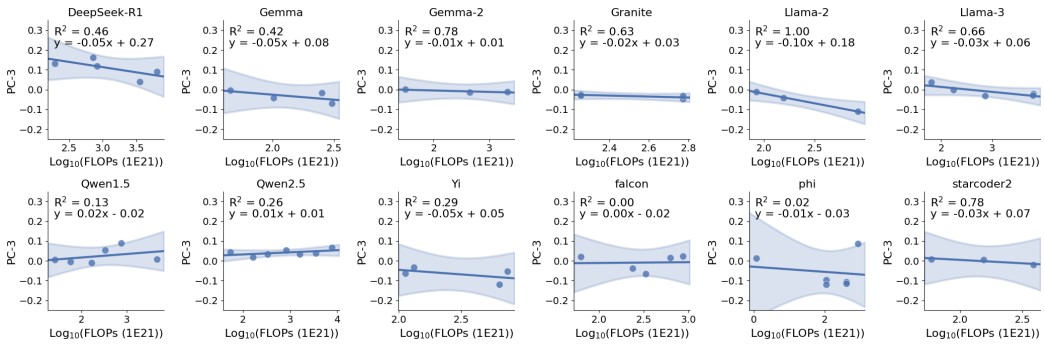

Figure 10: Linear relationship between the third principal component (PC-3) and log-compute across different model families.

## A.3 DETAIL CORRELATION VALUE BETWEEN OPENLLM LEADERBOARD AND EAI SKILLS

## A.4 IMPACT OF STRUCTURING OUTPUT

## A.5 OTHERS

add obscaling for goal interpretation on virtualhome and behavior

| Model | Family | Size (B) | Tokens (T) | FLOPs (1E21) | OpenLLM metric |
|---|---|---|---|---|---|
| Baichuan-7B | Baichuan | — | 1.20 | — | No |
| Baichuan2-7B-Base | Baichuan | 7 | 2.60 | 109.20 | No |
| Baichuan2-7B-Chat | Baichuan | 7 | 2.60 | 109.20 | No |
| DeepSeek-V3 | DeepSeek | 684.5 | 14.80 | 60783.60 | No |
| deepseek-coder-1.3b-base | DeepSeek-Coder | 1.3 | 2.00 | 15.60 | No |
| deepseek-coder-1.3b-instruct | DeepSeek-Coder | 1.3 | 2.00 | 15.60 | No |
| deepseek-coder-33b-base | DeepSeek-Coder | 33.3 | 2.00 | 396.00 | No |
| deepseek-coder-33b-instruct | DeepSeek-Coder | 33.3 | 2.00 | 399.60 | No |
| deepseek-coder-6.7b-base | DeepSeek-Coder | 6.7 | 2.00 | 80.40 | No |
| deepseek-coder-6.7b-instruct | DeepSeek-Coder | 6.7 | 2.00 | 80.40 | No |
| deepseek-coder-7b-base-v1.5 | DeepSeek-Coder | 6.9 | 2.00 | 82.80 | No |
| deepseek-coder-7b-instruct-v1.5 | DeepSeek-Coder | 6.9 | 2.00 | 82.80 | No |
| DeepSeek-R1 | DeepSeek-R1 | 684.5 | 14.80 | 60783.60 | No |
| DeepSeek-R1-Distill-Llama-70B | DeepSeek-R1 | 70.6 | 15.00 | 6354.00 | Yes |
| DeepSeek-R1-Distill-Llama-8B | DeepSeek-R1 | 8 | 15.00 | 720.00 | Yes |
| DeepSeek-R1-Distill-Qwen-1.5B | DeepSeek-R1 | 1.8 | 18.00 | 194.40 | Yes |
| DeepSeek-R1-Distill-Qwen-14B | DeepSeek-R1 | 14.8 | 18.00 | 1598.40 | Yes |
| DeepSeek-R1-Distill-Qwen-32B | DeepSeek-R1 | 32.8 | 18.00 | 3542.40 | Yes |
| DeepSeek-R1-Distill-Qwen-7B | DeepSeek-R1 | 7.6 | 18.00 | 820.80 | Yes |
| EXAONE-3.5-32B-Instruct | Exaone | 32 | 6.50 | 1248.00 | Yes |
| EXAONE-Deep-32B | Exaone | 32 | 6.50 | 1248.00 | No |
| gpt-oss-120b | GPT-OSS | 120.4 | — | — | No |
| gpt-oss-20b | GPT-OSS | 21.5 | — | — | No |
| gemma-1.1-2b-it | Gemma | 2.5 | 3.00 | 45.00 | Yes |
| gemma-1.1-7b-it | Gemma | 8.5 | 6.00 | 306.00 | Yes |
| gemma-7b | Gemma | 8.5 | 6.00 | 252.00 | Yes |
| gemma-7b-it | Gemma | 8.5 | 2.00 | 102.00 | Yes |
| gemma-2-27b | Gemma-2 | 27.2 | 13.00 | 2121.60 | Yes |
| gemma-2-27b-it | Gemma-2 | 27.2 | 13.00 | 2121.60 | Yes |
| gemma-2-2b | Gemma-2 | 2.6 | 2.00 | 31.20 | Yes |
| gemma-2-2b-it | Gemma-2 | 2.6 | 2.00 | 31.20 | Yes |
| gemma-2-9b | Gemma-2 | 9.2 | 8.00 | 441.60 | Yes |
| gemma-2-9b-it | Gemma-2 | 9.2 | 8.00 | 441.60 | Yes |
| gemma-2b | Gemma-2 | 2.5 | 6.00 | 72.00 | Yes |
| gemma-2b-it | Gemma-2 | 2.5 | 6.00 | 90.00 | Yes |
| gemma-3-12b-it | Gemma-3 | 12.2 | 12.00 | 878.40 | No |
| gemma-3-12b-pt | Gemma-3 | 12.2 | 12.00 | 878.40 | No |
| gemma-3-27b-it | Gemma-3 | 27.4 | 14.00 | 2301.60 | No |
| gemma-3-4b-it | Gemma-3 | 4.3 | 4.00 | 103.20 | No |
| gemma-3-4b-pt | Gemma-3 | 4.3 | 4.00 | 103.20 | No |
| granite-3.1-2b-base | Granite | 2.5 | 12.00 | 180.00 | Yes |
| granite-3.1-2b-instruct | Granite | 2.5 | 12.00 | 180.00 | Yes |
| granite-3.1-8b-base | Granite | 8.2 | 12.00 | 590.40 | Yes |
| granite-3.1-8b-instruct | Granite | 8.2 | 12.00 | 590.40 | Yes |
| granite-3.2-2b-instruct | Granite | 2.5 | 12.00 | 180.00 | Yes |
| granite-3.2-8b-instruct | Granite | 8.2 | 12.00 | 590.40 | Yes |
| granite-3.3-2b-base | Granite | 2.5 | 12.00 | 180.00 | No |
| granite-3.3-2b-instruct | Granite | 2.5 | 12.00 | 180.00 | No |
| granite-3.3-8b-base | Granite | 8.2 | 12.00 | 590.40 | No |
| granite-3.3-8b-instruct | Granite | 8.2 | 12.00 | 590.40 | No |
| Kimi-K2-Instruct | Kimi | 1000 | 15.50 | 93000.00 | No |
| Llama-4-Maverick-17B-128E-Instruct-FP8 | Llama | 401.6 | 22.00 | 53011.20 | No |
| Llama-4-Scout-17B-16E-Instruct | Llama | 108.6 | 40.00 | 26064.00 | No |
| llama3_8B_o4-mini-2025-04-16 | Llama | — | — | — | No |
| Llama-2-13b-hf | Llama-2 | 13 | 2.00 | 156.00 | Yes |
| Llama-2-70b-hf | Llama-2 | 69 | 2.00 | 840.00 | Yes |
| Llama-2-7b-hf | Llama-2 | 6.7 | 2.00 | 84.00 | Yes |
| Llama-3.1-70B | Llama-3 | 70.6 | 15.00 | 6354.00 | Yes |
| Llama-3.2-1B | Llama-3 | 1.2 | 9.00 | 64.80 | Yes |
| Llama-3.2-3B | Llama-3 | 3.2 | 9.00 | 172.80 | Yes |
| Llama-3.3-70B-Instruct | Llama-3 | 70.6 | 15.00 | 6354.00 | Yes |
| Meta-Llama-3-70B | Llama-3 | 70.6 | 15.00 | 6300.00 | Yes |
| Meta-Llama-3-70B-Instruct | Llama-3 | 70.6 | 15.00 | 6354.00 | Yes |
| Meta-Llama-3-8B | Llama-3 | 8 | 15.00 | 720.00 | Yes |
| Meta-Llama-3-8B-Instruct | Llama-3 | 8 | 15.00 | 720.00 | Yes |

Table 1: Model summary (part 1 of 2). Models sorted by family then name; OpenLLM metric = non-NA 'Average'.

| Model | Family | Size (B) | Tokens (T) | FLOPs (1E21) | OpenLLM metric |
|---|---|---|---|---|---|
| Mistral-7B-Instruct-v0.2 | Mistral | 7.2 | — | — | Yes |
| Mixtral-8x7B-Instruct-v0.1 | Mistral | 46.7 | — | — | Yes |
| Qwen-14B | Qwen | 14.2 | 3.00 | 252.00 | No |
| Qwen-72B | Qwen | 72.3 | 3.00 | 1296.00 | No |
| Qwen-7B | Qwen | 7.7 | 2.40 | 100.80 | No |
| Qwen1.5-1.8B | Qwen1.5 | 1.8 | 2.40 | 25.92 | Yes |
| Qwen1.5-110B | Qwen1.5 | 111.2 | 7.00 | 4670.40 | Yes |
| Qwen1.5-14B | Qwen1.5 | 14.2 | 4.00 | 336.00 | Yes |
| Qwen1.5-32B | Qwen1.5 | 32.5 | 4.00 | 768.00 | Yes |
| Qwen1.5-4B | Qwen1.5 | 4 | 2.40 | 57.60 | Yes |
| Qwen1.5-72B | Qwen1.5 | 72.3 | 3.00 | 1296.00 | No |
| Qwen1.5-7B | Qwen1.5 | 7.7 | 4.00 | 168.00 | Yes |
| Qwen2.5-0.5B | Qwen2.5 | 0.5 | 18.00 | 54.00 | Yes |
| Qwen2.5-1.5B | Qwen2.5 | 1.5 | 18.00 | 162.00 | Yes |
| Qwen2.5-14B | Qwen2.5 | 14.8 | 18.00 | 1598.40 | Yes |
| Qwen2.5-32B | Qwen2.5 | 32.8 | 18.00 | 3542.40 | Yes |
| Qwen2.5-3B | Qwen2.5 | 3.1 | 18.00 | 334.80 | Yes |
| Qwen2.5-72B | Qwen2.5 | 72.7 | 18.00 | 7851.60 | Yes |
| Qwen2.5-7B | Qwen2.5 | 7.6 | 18.00 | 820.80 | Yes |
| Qwen3-0.6B | Qwen3 | 0.8 | 36.00 | 172.80 | No |
| Qwen3-1.7B | Qwen3 | 2 | 36.00 | 432.00 | No |
| Qwen3-14B | Qwen3 | 14.8 | 36.00 | 3196.80 | No |
| Qwen3-235B-A22B-Thinking-2507 | Qwen3 | 235.1 | 36.00 | 50781.60 | No |
| Qwen3-32B | Qwen3 | 32.8 | 36.00 | 7084.80 | No |
| Qwen3-4B | Qwen3 | 4 | 36.00 | 864.00 | No |
| Qwen3-8B | Qwen3 | 8.2 | 36.00 | 1771.20 | No |
| Yi-1.5-34B | Yi | 34.4 | 3.60 | 743.04 | Yes |
| Yi-1.5-34B-Chat | Yi | 34.4 | 3.60 | 743.04 | Yes |
| Yi-1.5-6B | Yi | 6.1 | 3.60 | 131.76 | Yes |
| Yi-1.5-6B-Chat | Yi | 6.1 | 3.60 | 131.76 | Yes |
| Yi-1.5-9B | Yi | 8.8 | 3.60 | 190.08 | Yes |
| Yi-34B | Yi | 34.4 | 3.10 | 639.84 | Yes |
| Yi-6B | Yi | 6.1 | 3.10 | 113.46 | Yes |
| Yi-Coder-1.5B | Yi | 1.5 | 2.40 | 21.60 | No |
| Yi-Coder-1.5B-Chat | Yi | 1.5 | 2.40 | 21.60 | No |
| Yi-Coder-9B | Yi | 8.8 | 2.40 | 126.72 | No |
| Yi-Coder-9B-Chat | Yi | 8.8 | 2.40 | 126.72 | Yes |
| Falcon3-10B-Base | falcon | 10.3 | 14.00 | 865.20 | Yes |
| Falcon3-7B-Base | falcon | 7.5 | 14.00 | 630.00 | Yes |
| falcon-11B | falcon | 11.1 | 5.00 | 333.00 | Yes |
| falcon-40b | falcon | 41.8 | 1.00 | 240.00 | Yes |
| falcon-7b | falcon | 7.2 | 1.50 | 63.00 | Yes |
| gpt-4.1-2025-04-14 | gpt-4.1-2025-04-14 | — | — | — | No |
| gpt-4.1-mini-2025-04-14 | gpt-4.1-mini-2025-04-14 | — | — | — | No |
| gpt-4.1-nano-2025-04-14 | gpt-4.1-nano-2025-04-14 | — | — | — | No |
| o4-mini-2025-04-16 | o4-mini-2025-04-16 | — | — | — | No |
| Phi-3-medium-128k-instruct | phi | 14 | 4.80 | 403.20 | Yes |
| Phi-3-medium-4k-instruct | phi | 14 | 4.80 | 403.20 | Yes |
| Phi-3-mini-128k-instruct | phi | 3.8 | 4.90 | 111.72 | Yes |
| Phi-3-mini-4k-instruct | phi | 3.8 | 4.90 | 111.72 | Yes |
| phi-1_5 | phi | 1.4 | 0.15 | 1.17 | Yes |
| phi-4 | phi | 14.7 | 9.80 | 864.36 | Yes |
| starcoderbase | starcoder | 15.5 | 1.00 | 93.00 | No |
| starcoderbase-1b | starcoder | 15.5 | 1.00 | 6.00 | No |
| starcoderbase-3b | starcoder | 15.5 | 1.00 | 18.00 | No |
| starcoderbase-7b | starcoder | 15.5 | 1.00 | 42.00 | No |
| starcoder2-15b | starcoder2 | 16 | 4.30 | 387.00 | Yes |
| starcoder2-3b | starcoder2 | 3 | 3.30 | 59.40 | Yes |
| starcoder2-7b | starcoder2 | 7.2 | 3.70 | 155.40 | Yes |

Table 2: Model summary (part 2 of 2). Models sorted by family then name; OpenLLM metric = non-NA 'Average'.

Table 3: Correlation between Base LLM Benchmarks and Virtualhome Action Sequencing Task Performance. Bold values indicate strong correlations ($|r| \geq 0.7$), italic values indicate moderate correlations ($0.5 \leq |r| < 0.7$).

| EAI Task Metrics | GPQA | MUSR | IFEval | MMLU-PRO | BBH | MATH Lvl 5 |
|---|---|---|---|---|---|---|
| Task Success | *0.525* | *0.558* | *0.618* | **0.714** | **0.754** | **0.782** |
| State Goal | *0.554* | *0.564* | *0.589* | **0.706** | **0.742** | **0.761** |
| Relation Goal | *0.521* | *0.558* | *0.577* | **0.707** | **0.743** | **0.783** |
| Action Goal | *0.514* | *0.531* | *0.622* | **0.704** | **0.746** | **0.779** |
| Total Goal | *0.552* | *0.571* | *0.608* | **0.725** | **0.764** | **0.792** |
| Execution Success | *0.507* | *0.532* | *0.648* | **0.701** | **0.747** | **0.773** |
| Parsing | 0.480 | 0.382 | *0.530* | *0.535* | *0.574* | 0.496 |
| Hallucination | 0.081 | 0.212 | 0.152 | 0.217 | 0.217 | 0.227 |
| Predicate Arg | -0.204 | -0.085 | -0.345 | -0.308 | -0.382 | -0.080 |
| Wrong Order | -0.369 | -0.350 | -0.016 | -0.485 | -0.388 | -0.417 |
| Missing Step | -0.360 | -0.329 | -0.386 | -0.355 | -0.380 | -0.255 |
| Affordance | -0.198 | -0.102 | -0.189 | -0.199 | -0.205 | -0.122 |
| Additional Step | -0.317 | -0.286 | 0.112 | -0.304 | -0.277 | -0.190 |

Table 4: Correlation between Base LLM Benchmarks and Behavior Action Sequencing Task Performance. Bold values indicate strong correlations ($|r| \geq 0.7$), italic values indicate moderate correlations ($0.5 \leq |r| < 0.7$).

| EAI Task Metrics | GPQA | MUSR | IFEval | MMLU-PRO | BBH | MATH Lvl 5 |
|---|---|---|---|---|---|---|
| Task Success | 0.311 | 0.203 | *0.689* | *0.601* | *0.613* | *0.604* |
| State Goal | 0.282 | 0.264 | **0.702** | *0.613* | *0.581* | *0.649* |
| Relation Goal | 0.355 | 0.148 | **0.709** | *0.526* | *0.620* | *0.510* |
| Total Goal | 0.340 | 0.196 | **0.740** | *0.578* | *0.629* | *0.589* |
| Execution Success | 0.333 | 0.184 | **0.726** | *0.587* | *0.615* | *0.574* |
| Parsing | 0.081 | 0.062 | **0.838** | 0.307 | 0.372 | 0.295 |
| Hallucination | 0.137 | -0.035 | -0.187 | 0.120 | 0.111 | 0.225 |
| Predicate Arg | 0.201 | 0.016 | -0.112 | 0.256 | 0.220 | 0.247 |
| Wrong Order | -0.171 | -0.131 | **-0.816** | -0.356 | -0.469 | -0.444 |
| Missing Step | -0.010 | 0.069 | **-0.783** | -0.229 | -0.296 | -0.254 |
| Additional Step | 0.063 | 0.027 | *-0.515* | -0.147 | -0.180 | -0.208 |

Table 5: Correlation between Base LLM Benchmarks and Virtualhome Goal Interpretation Task Performance. Bold values indicate strong correlations ($|r| \geq 0.7$), italic values indicate moderate correlations ($0.5 \leq |r| < 0.7$).

| EAI Task Metrics | GPQA | MUSR | IFEval | MMLU-PRO | BBH | MATH Lvl 5 |
|---|---|---|---|---|---|---|
| Node F1 | 0.495 | 0.237 | 0.087 | 0.372 | 0.459 | 0.136 |
| Edge F1 | *0.531* | 0.305 | 0.163 | 0.456 | *0.567* | 0.274 |
| Action F1 | 0.339 | 0.175 | 0.314 | 0.250 | 0.343 | 0.163 |
| All F1 | *0.554* | 0.289 | 0.162 | 0.427 | *0.526* | 0.187 |

Table 6: Correlation between Base LLM Benchmarks and Behavior Goal Interpretation Task Performance. Bold values indicate strong correlations ($|r| \geq 0.7$), italic values indicate moderate correlations ($0.5 \leq |r| < 0.7$).

| EAI Task Metrics | GPQA | MUSR | IFEval | MMLU-PRO | BBH | MATH Lvl 5 |
|---|---|---|---|---|---|---|
| Overall F1 | *0.627* | *0.642* | 0.484 | **0.821** | **0.742** | **0.761** |
| State Goal F1 | *0.576* | *0.605* | 0.459 | **0.777** | **0.707** | **0.780** |
| Relation Goal F1 | *0.646* | *0.652* | *0.500* | **0.837** | **0.765** | **0.727** |
| State Hallucinati... | 0.309 | 0.300 | 0.402 | 0.493 | 0.403 | 0.466 |
| Object Hallucinat... | 0.325 | 0.324 | 0.420 | *0.511* | 0.422 | 0.490 |
| Format Error Rate | 0.295 | 0.280 | 0.419 | 0.473 | 0.384 | 0.460 |
| Grammatically Val... | -0.446 | -0.412 | *-0.515* | *-0.634* | *-0.518* | *-0.574* |

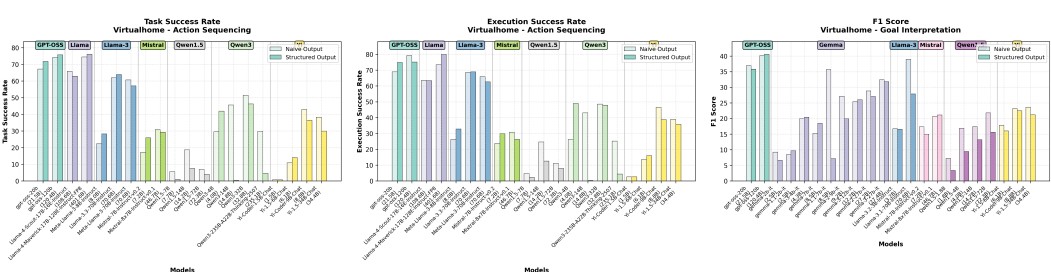

Figure 11: Comparison of observational scaling laws for standard generation (Base Model) versus structured decoding (Model with Decoder Masking) on the Virtualhome goal interpretation task. The plots show performance on action sequencing and goal interpretation tasks on Virtualhome.

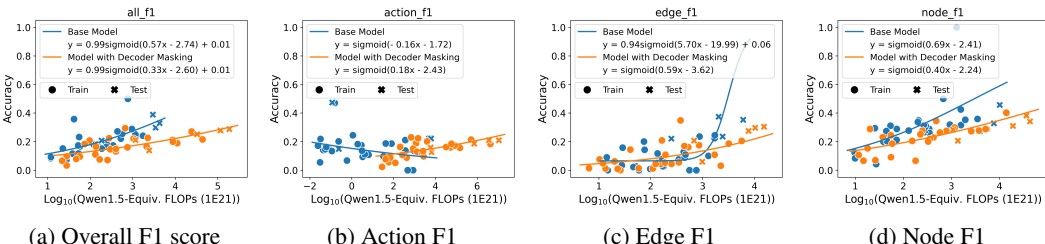

(a) Overall F1 score      (b) Action F1      (c) Edge F1      (d) Node F1

Figure 12: Comparison of observational scaling laws for standard generation (Base Model) versus structured decoding (Model with Decoder Masking) on the Virtualhome goal interpretation task. The plots show performance on four different F1 metrics as a function of model scale. While overall performance is comparable, structured decoding significantly degrades scaling performance on granular sub-tasks, particularly for Edge F1, suggesting that output constraints can hinder the learning of complex relational structures.

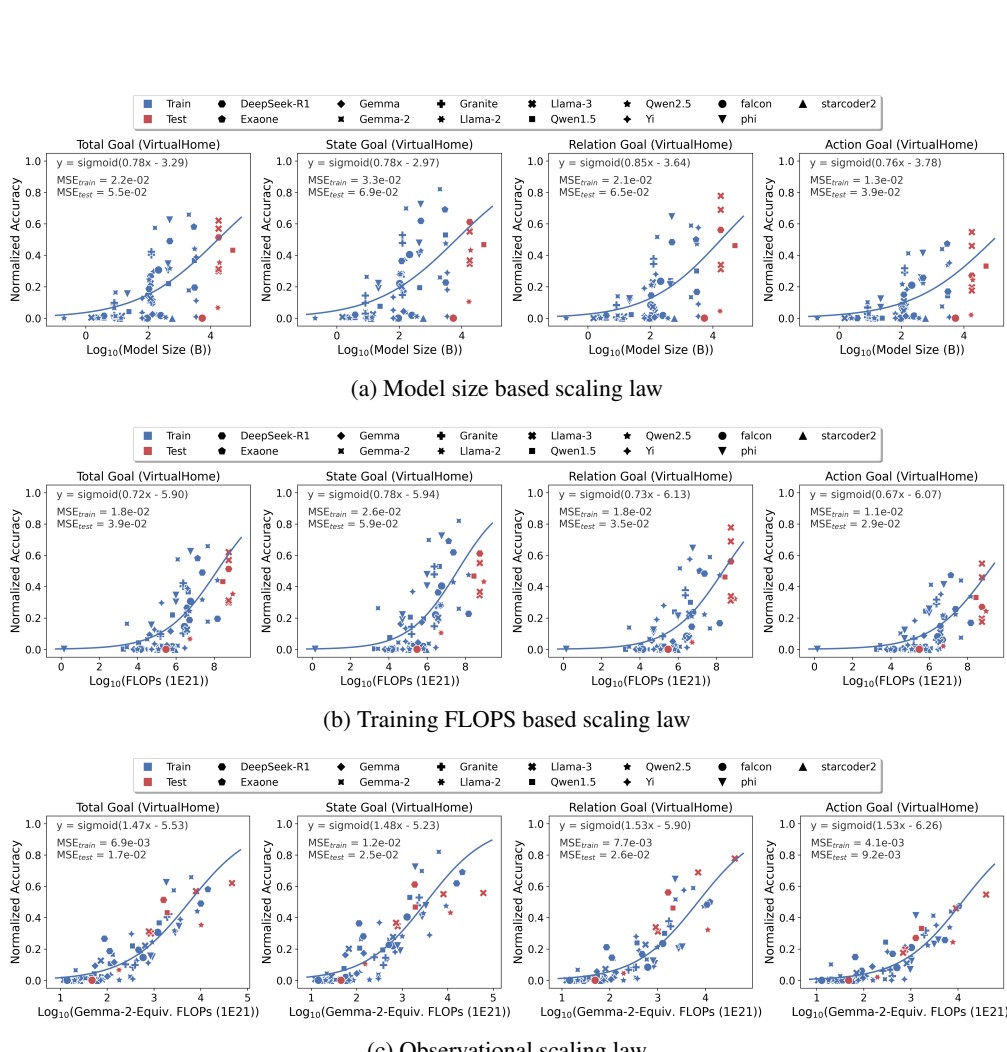

(a) Model size based scaling law

(b) Training FLOPS based scaling law

(c) Observational scaling law

Figure 13: Scaling curves of action sequencing on Vritualhome.

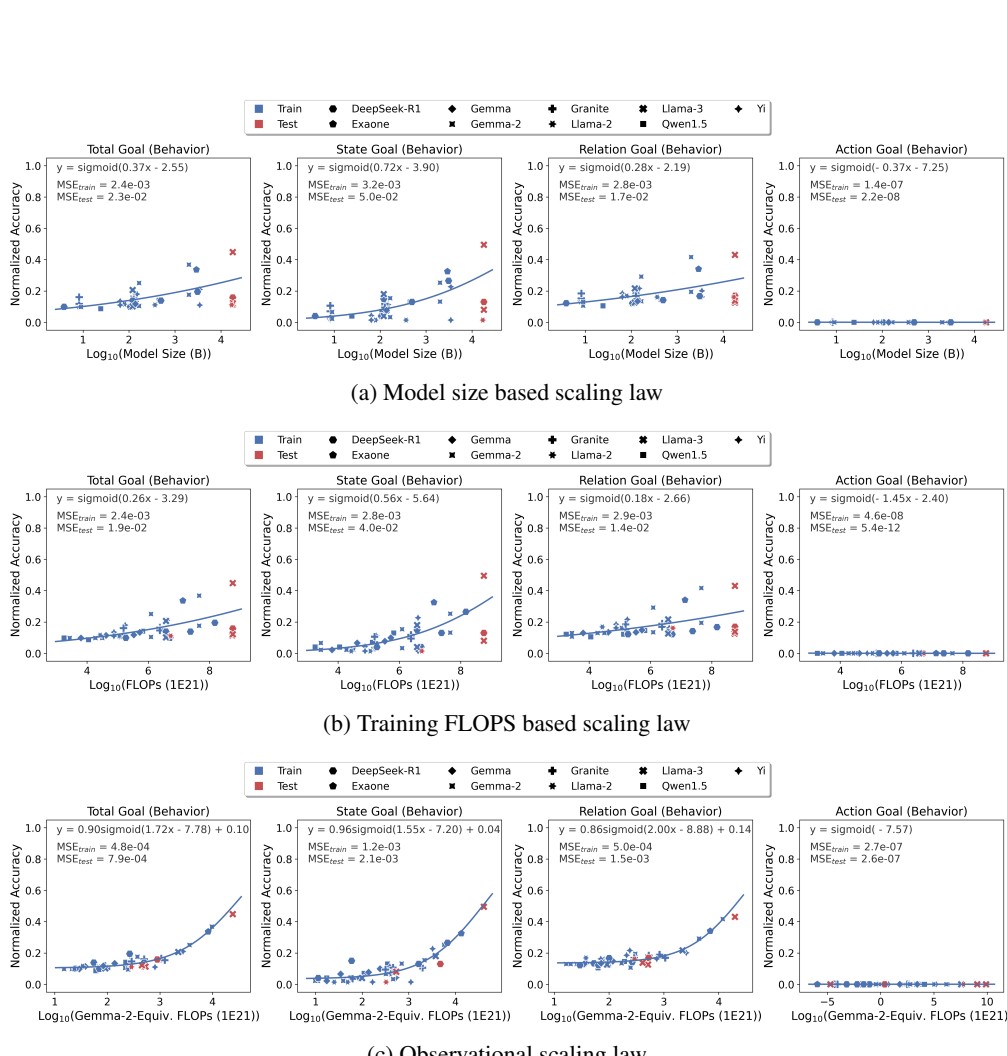

(a) Model size based scaling law

(b) Training FLOPS based scaling law

(c) Observational scaling law

Figure 14: Scaling curves of action sequencing on Behavior. Action goal is all 0 in behacior simualtion.

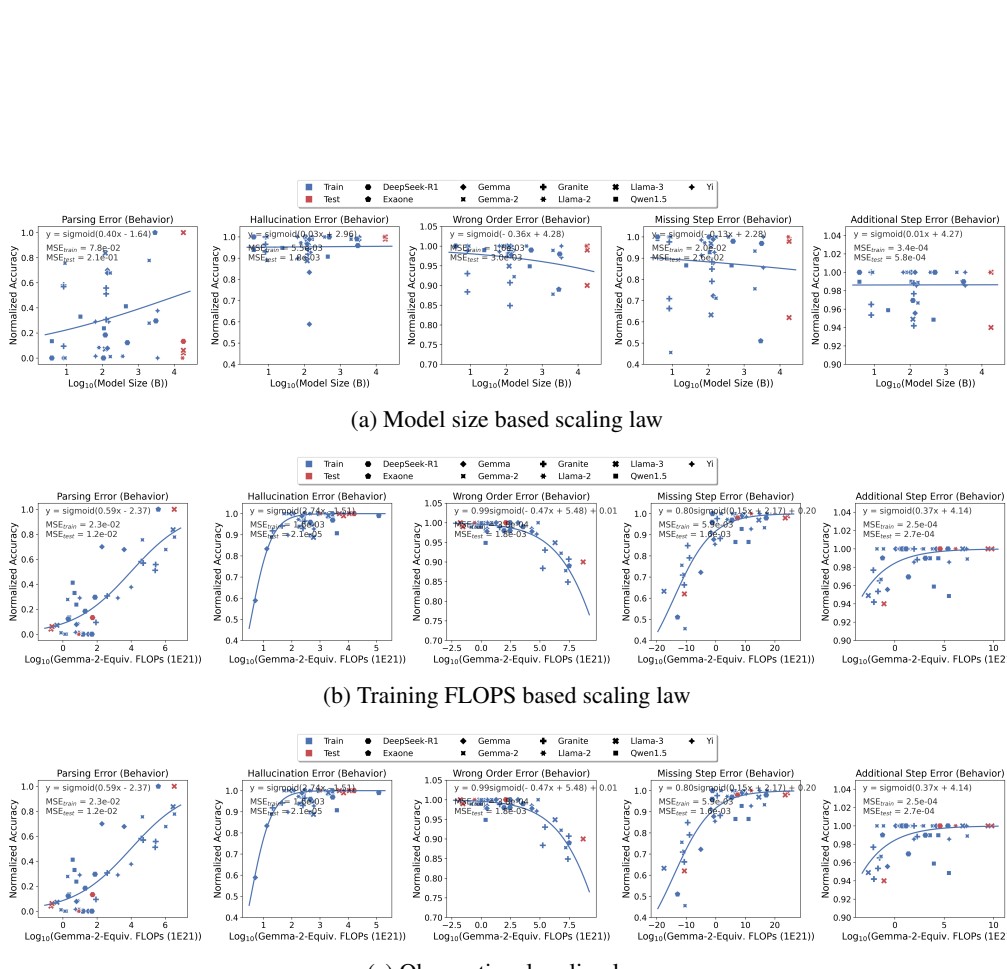

(a) Model size based scaling law

(b) Training FLOPS based scaling law

(c) Observational scaling law

Figure 15: Scaling curves of action sequencing on Behavior.

