# OpenReview forum: "Observational Scaling Laws in LLM-based Embodied Decision Making"
_ICLR.cc/2026/Conference — ICLR 2026 Conference Withdrawn Submission_

### Official Review · Reviewer_RaeT · 2025-10-31

**Soundness:** 3
**Presentation:** 3
**Contribution:** 1
**Rating:** 2
**Confidence:** 4

**Summary:**

This paper applies the observational scaling law framework to embodied decision-making tasks via the Embodied Agent Interface. Instead of training models at multiple scales, the authors estimate scaling trends across 125 publicly available LLMs and attempt to predict performance on downstream embodied skills, quantify simulation-to-simulation gaps, and measure the effect of structured decoding.

**Strengths:**

- The study on embodied benchmarks is thorough, covering a diverse family of LLMs across scales on different embodied benchmarks, and observes idiosyncratic performance change when structured decoding is used to ensure valid format in generating plans and actions.
- The paper is clearly written, the visualizations are clear and the experiments appear to be carefully executed.

**Weaknesses:**

- The novelty is very limited. The core method originates from observational scaling law. The contribution here is to apply and validate it to a different benchmark suite. I don't see any major contributions other than empirically fitting an existing framework to embodied benchmarks with the conclusion that this existing methodology works for certain benchmarks that the original authors did not evaluate. Therefore, I am only convinced on the second contribution that the authors mentioned -- analyzing gaps between different simulation environments and structured outputs that are domain-specific. But I don't believe this contribution alone is suitable for a conference paper.
- Even only considering validating observational scaling law on decision-making tasks as a novelty, it is still very incremental because observational scaling law already discussed scaling on agentic tasks, and the evaluated benchmarks in this work only have a different domain (planning and execution in physical world tasks vs digital agents e.g., coding).

**Questions:**

- Can the authors clearly articulate what new conceptual or methodological insight is introduced beyond applying observational scaling to EAI?
- How does EAI differ from AgentBench/AgentBoard in terms of scaling behavior? Are there qualitative failure modes unique to embodied settings?
- Could the authors analyze where the scaling law breaks down in embodied tasks? That could provide deeper value.

---

### Official Review · Reviewer_sDo5 · 2025-10-31

**Soundness:** 2
**Presentation:** 3
**Contribution:** 3
**Rating:** 6
**Confidence:** 3

**Summary:**

This paper introduces a novel observational method for deriving scaling laws for Large Language Models (LLMs) in embodied decision-making tasks. Instead of the conventional, resource-intensive approach of training models from scratch at various scales, the authors leverage a large set of 125 publicly available pre-trained LLMs. The core idea is to model performance not as a direct function of compute, but as a function of a low-dimensional "capability space" derived from standard NLP benchmarks via Principal Component Analysis (PCA). This generalized framework allows for unifying scaling trends across diverse model families (e.g., Llama, Qwen, Gemma).
The authors validate their approach on the Embodied Agent Interface (EAI) benchmark, demonstrating that their observational scaling law achieves significantly higher predictive accuracy (over 50% improvement in MSE) for larger models (>40B parameters) compared to traditional laws based on model size or training FLOPs.

**Strengths:**

1. The claims are well-supported by extensive experiments on 125 open LLMs from 28 different families. The results clearly show that the proposed observational scaling law significantly outperforms traditional compute-based baselines in extrapolating performance to larger models, as evidenced by the substantially lower test MSE.
2. The paper is well-written, and the methodology is explained clearly. The figures, particularly the comparisons of scaling curves in Figure 4 and the analysis of regression coefficients in Figure 5, are effective at conveying the key results and their implications.

**Weaknesses:**

- Limited Scope of Models: The study is confined to open-source models. While this is a practical necessity for the methodology, it overlooks the performance of state-of-the-art, closed-source models (e.g., from the GPT-4 or Claude families), which often define the upper frontier of capabilities. It remains an open question whether these models would conform to the same scaling laws, and their exclusion might limit the generality of the conclusions.
- Generalizability to New Tasks and Paradigms: The scaling laws are derived and validated on the specific tasks within the EAI benchmark. The paper does not provide evidence on how well these laws might generalize to entirely new embodied tasks or different simulation environments. Furthermore, the current landscape of LLMs is largely based on the transformer architecture. If a fundamentally new training paradigm or model architecture emerges, it is uncertain whether this observational framework would remain effective without significant recalibration.

**Questions:**

- The paper sets the number of principal components to K=3, noting it captures ~97% of the variance. How was this value of K chosen? Was there a more principled method used, or was it based on this explained variance threshold? How sensitive are the final predictions and the interpretation of the capability space to different choices of K (e.g., K=2 or K=5)?

- As noted in the weaknesses, the framework's reliance on current benchmarks and model architectures raises questions about its long-term robustness. Can the authors comment on how they envision this method adapting to (a) novel embodied tasks with different challenges, and (b) potential future shifts in LLM training paradigms that might alter the relationship between standard benchmarks and embodied skills?

- The predictive accuracy of the scaling laws is evaluated using Mean Squared Error (MSE). While MSE is a standard metric, it can be sensitive to outliers. Is one MSE metric enough? Have the authors considered alternative metrics, such as Mean Absolute Error (MAE) or a ranking correlation metric (like Spearman's ρ), to provide a more robust assessment of the model's predictive power?


- The paper reports a strong correlation (R² > 0.89) for the log-linear fit between capability (PC-1) and compute within model families. A high R² on observed data indicates a good fit but does not guarantee strong extrapolative performance. Could the authors elaborate on the risk of this relationship being a good interpolation that doesn't hold for models significantly larger than those in the training set?

- The methodology filters out models with "zero task performance" or "missing benchmark scores." This could introduce a selection bias, as these excluded models (especially the zero-performance ones) might represent a critical low-capability regime. Could the exclusion of these data points potentially skew the fitted scaling curve and affect the model's ability to accurately predict the initial "emergence" of a capability?

---

### Official Review · Reviewer_hjGC · 2025-11-01

**Soundness:** 3
**Presentation:** 3
**Contribution:** 3
**Rating:** 4
**Confidence:** 4

**Summary:**

This paper proposes an observational scaling framework to derive scaling laws for LLM performance in embodied decision-making tasks without training new models, using a low-dimensional capability space to unify trends across 28 model families (125 LLMs total). Validated on the Embodied Agent Interface (EAI) benchmark, it predicts emergent skills in models >40B from smaller ones, quantifies performance gaps between simulators, and measures degradation from structured decoding, achieving 50% better predictive accuracy than compute-based laws.

**Strengths:**

1.Novel observational approach bypasses costly retraining by leveraging existing LLMs and upstream benchmarks, enabling unified scaling across heterogeneous families like LLaMA, Qwen, and Gemma.
2.Strong empirical validation on EAI with 125 models shows high predictive power, e.g., forecasting large-model performance from <40B data and quantifying sim-to-real gaps.
3.Practical insights into interventions: reveals structured outputs degrade performance and simulator differences impact scaling, with clear visualizations and ablations.
4.Theoretically grounded in generalized scaling (log-linear trends in capability space), bridging language and embodied domains.

**Weaknesses:**

1.Relies on upstream benchmarks (e.g., reasoning, coding) as proxies for capabilities, which may not fully capture embodied-specific skills like spatial reasoning or dynamics modeling.
2.Limited to open LLMs up to 40B-70B scale; excludes proprietary giants like GPT-4, potentially biasing cross-family generalizations.
3.EAI benchmark is simulation-only; no real-world validation, so sim gap quantification remains hypothetical without physical robot tests.
4.Predictive accuracy claims (50% over baselines) lack error bars or sensitivity analyses to noisy upstream data or family-specific efficiencies.
5.Focuses on zero-shot LLM prompting; ignores fine-tuning or VLA integrations that could alter scaling behaviors.

**Questions:**

1.How do you select the low-dimensional capability space? Any ablation on including/excluding specific upstream benchmarks?
2.Can the framework predict scaling for proprietary models (e.g., GPT-5) if upstream scores are available?
3.What are the compute costs for evaluating 125 LLMs on EAI—e.g., total inference tokens or time?
4.How robust is the sim gap quantification to different EAI variants or non-household tasks?
5.Does the degradation from structured decoding hold for advanced techniques like JSON mode or tool-calling?

---

### Note · Authors · 2026-01-08

**Comment:**

This paper was withdrawn because the submission was not approved by PIs.

**Withdrawal Confirmation:**

I have read and agree with the venue's withdrawal policy on behalf of myself and my co-authors.